# Natural Neural Networks

**Guillaume Desjardins, Karen Simonyan, Razvan Pascanu, Koray Kavukcuoglu**
{gdesjardins,simonyan,razp,korayk}@google.com
Google DeepMind, London

## Abstract

We introduce Natural Neural Networks, a novel family of algorithms that speed up convergence by adapting their internal representation during training to improve conditioning of the Fisher matrix. In particular, we show a specific example that employs a simple and efficient reparametrization of the neural network weights by implicitly whitening the representation obtained at each layer, while preserving the feed-forward computation of the network. Such networks can be trained efficiently via the proposed Projected Natural Gradient Descent algorithm (PRONG), which amortizes the cost of these reparametrizations over many parameter updates and is closely related to the Mirror Descent online learning algorithm. We highlight the benefits of our method on both unsupervised and supervised learning tasks, and showcase its scalability by training on the large-scale ImageNet Challenge dataset.

## 1 Introduction

Deep networks have proven extremely successful across a broad range of applications. While their deep and complex structure affords them a rich modeling capacity, it also creates complex dependencies between the parameters which can make learning difficult via first order stochastic gradient descent (SGD). As long as SGD remains the workhorse of deep learning, our ability to extract high-level representations from data may be hindered by difficult optimization, as evidenced by the boost in performance offered by batch normalization (BN) [7] on the Inception architecture [25].

Though its adoption remains limited, the natural gradient [1] appears ideally suited to these difficult optimization issues. By following the direction of steepest descent on the probabilistic manifold, the natural gradient can make constant progress over the course of optimization, as measured by the Kullback-Leibler (KL) divergence between consecutive iterates. Utilizing the proper distance measure ensures that the natural gradient is invariant to the parametrization of the model. Unfortunately, its application has been limited due to its high computational cost. Natural gradient descent (NGD) typically requires an estimate of the Fisher Information Matrix (FIM) which is square in the number of parameters, and worse, it requires computing its inverse. Truncated Newton methods can avoid explicitly forming the FIM in memory [12, 15], but they require an expensive iterative procedure to compute the inverse. Such computations can be wasteful as they do not take into account the highly structured nature of deep models.

Inspired by recent work on model reparametrizations [17, 13], our approach starts with a simple question: can we devise a neural network architecture whose Fisher is constrained to be identity? This is an important question, as SGD and NGD would be equivalent in the resulting model. The main contribution of this paper is in providing a simple, theoretically justified network reparametrization which approximates via first-order gradient descent, a block-diagonal natural gradient update over layers. Our method is computationally efficient due to the local nature of the reparametrization, based on whitening, and the amortized nature of the algorithm. Our second contribution is in unifying many heuristics commonly used for training neural networks, under the roof of the natural gradient, while highlighting an important connection between model reparametrizations and Mirror Descent [3]. Finally, we showcase the efficiency and the scalability of our method

across a broad-range of experiments, scaling our method from standard deep auto-encoders to large convolutional models on ImageNet[20], trained across multiple GPUs. This is to our knowledge the first-time a (non-diagonal) natural gradient algorithm is scaled to problems of this magnitude.

## 2   The Natural Gradient

This section provides the necessary background and derives a particular form of the FIM whose structure will be key to our efficient approximation. While we tailor the development of our method to the classification setting, our approach generalizes to regression and density estimation.

### 2.1   Overview

We consider the problem of fitting the parameters $\theta \in \mathbb{R}^N$ of a model $p(y \mid x; \theta)$ to an empirical distribution $\pi(x, y)$ under the log-loss. We denote by $x \in \mathcal{X}$ the observation vector and $y \in \mathcal{Y}$ its associated label. Concretely, this stochastic optimization problem aims to solve:

$$\theta^* \quad \in \quad \text{argmin}_\theta \, \mathbb{E}_{(x,y)\sim\pi} \left[ -\log p(y \mid x, \theta) \right]. \tag{1}$$

Defining the per-example loss as $\ell(x, y)$, Stochastic Gradient Descent (SGD) performs the above minimization by iteratively following the direction of steepest descent, given by the column vector $\nabla = \mathbb{E}_\pi \left[ d\ell/d\theta \right]$. Parameters are updated using the rule $\theta^{(t+1)} \leftarrow \theta^{(t)} - \alpha^{(t)} \nabla^{(t)}$, where $\alpha$ is a learning rate. An equivalent proximal form of gradient descent [4] reveals the precise nature of $\alpha$:

$$\theta^{(t+1)} \quad = \quad \text{argmin}_\theta \left\{ \langle \theta, \nabla \rangle + \frac{1}{2\alpha^{(t)}} \left\| \theta - \theta^{(t)} \right\|_2^2 \right\} \tag{2}$$

Namely, each iterate $\theta^{(t+1)}$ is the solution to an auxiliary optimization problem, where $\alpha$ controls the distance between consecutive iterates, using an $L_2$ distance. In contrast, the natural gradient relies on the KL-divergence between iterates, a more appropriate distance measure for probability distributions. Its metric is determined by the Fisher Information matrix,

$$F_\theta = \mathbb{E}_{x \sim \pi} \left\{ \mathbb{E}_{y \sim p(y|x,\theta)} \left[ \left( \frac{\partial \log p}{\partial \theta} \right) \left( \frac{\partial \log p}{\partial \theta} \right)^T \right] \right\}, \tag{3}$$

i.e. the covariance of the gradients of the model log-probabilities wrt. its parameters. The natural gradient direction is then obtained as $\nabla_N = F_\theta^{-1} \nabla$. See [15, 14] for a recent overview of the topic.

### 2.2   Fisher Information Matrix for MLPs

We start by deriving the precise form of the Fisher for a canonical multi-layer perceptron (MLP) composed of $L$ layers. We consider the following deep network for binary classification, though our approach generalizes to an arbitrary number of output classes.

$$p(y = 1 \mid x) \equiv h_L \quad = \quad f_L(W_L h_{L-1} + b_L) \tag{4}$$
$$\cdots$$
$$h_1 \quad = \quad f_1 (W_1 x + b_1)$$

The parameters of the MLP, denoted $\theta = \{W_1, b_1, \cdots, W_L, b_L\}$, are the weights $W_i \in \mathbb{R}^{N_i \times N_{i-1}}$ connecting layers $i$ and $i - 1$, and the biases $b_i \in \mathbb{R}^{N_i}$. $f_i$ is an element-wise non-linear function.

Let us define $\delta_i$ to be the backpropagated gradient through the $i$-th non-linearity. We ignore the off block-diagonal components of the Fisher matrix and focus on the block $F_{W_i}$, corresponding to interactions between parameters of layer $i$. This block takes the form:

$$F_{W_i} = \mathbb{E}_{\substack{x \sim \pi \\ y \sim p}} \left[ \mathbf{vec} \left( \delta_i h_{i-1}^T \right) \mathbf{vec} \left( \delta_i h_{i-t}^T \right)^T \right],$$

where $\mathbf{vec}(X)$ is the vectorization function yielding a column vector from the *rows* of matrix $X$.

Assuming that $\delta_i$ and activations $h_{i-1}$ are independent random variables, we can write:

$$F_{W_i}(km, ln) \approx \mathbb{E}_{\substack{x \sim \pi \\ y \sim p}} \left[ \delta_i(k)\delta_i(l) \right] \mathbb{E}_\pi \left[ h_{i-1}(m)h_{i-1}(n) \right], \tag{5}$$

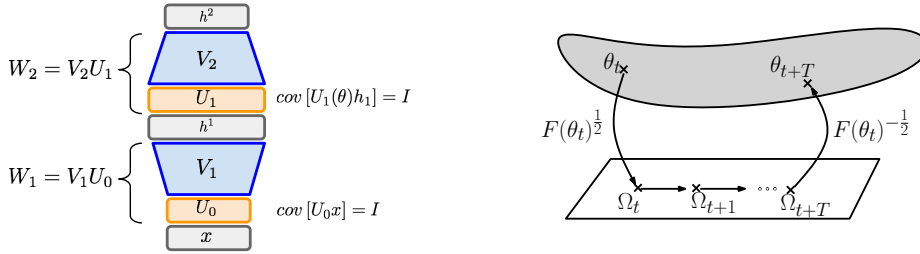

Figure 1: (a) A 2-layer natural neural network. (b) Illustration of the projections involved in PRONG.

where $X(i, j)$ is the element at row $i$ and column $j$ of matrix $X$ and $x(i)$ is the $i$-th element of vector $x$. $F_{W_i}(km, ln)$ is the entry in the Fisher capturing interactions between parameters $W_i(k, m)$ and $W_j(l, n)$. Our hypothesis, verified experimentally in Sec. 4.1, is that we can greatly improve conditioning of the Fisher by enforcing that $\mathbb{E}_\pi \left[ h_i h_i^T \right] = I$, for all layers of the network, despite ignoring possible correlations in the $\delta$'s and off block diagonal terms of the Fisher.

## 3   Projected Natural Gradient Descent

This section introduces Whitened Neural Networks (WNN), which perform approximate whitening of their internal representations. We begin by presenting a novel whitened neural layer, with the assumption that the network statistics $\mu_i(\theta) = \mathbb{E}[h_i]$ and $\Sigma_i(\theta) = \mathbb{E}[h_i h_i^T]$ are fixed. We then show how these layers can be adapted to efficiently track population statistics over the course of training. The resulting learning algorithm is referred to as Projected Natural Gradient Descent (PRONG). We highlight an interesting connection between PRONG and Mirror Descent in Section 3.3.

### 3.1   A Whitened Neural Layer

The building block of WNN is the following neural layer,

$$ h_i \quad = \quad f_i \left( V_i U_{i-1} \left( h_{i-1} - c_{i-1} \right) + d_i \right). \tag{6} $$

Compared to Eq. 4, we have introduced an explicit centering parameter $c_{i-1} \in \mathbb{R}^{N_{i-1}}$, equal to $\mu_{i-1}$, which ensures that the input to the dot product has zero mean in expectation. This is analogous to the centering reparametrization for Deep Boltzmann Machines [13]. The weight matrix $U_{i-1} \in \mathbb{R}^{N_{i-1} \times N_{i-1}}$ is a per-layer PCA-whitening matrix whose rows are obtained from an eigendecomposition of $\Sigma_{i-1}$:

$$ \Sigma_i = \tilde{U}_i \cdot diag \left( \lambda_i \right) \cdot \tilde{U}_i^T \implies U_i = diag \left( \lambda_i + \epsilon \right)^{-\frac{1}{2}} \cdot \tilde{U}_i^T. \tag{7} $$

The hyper-parameter $\epsilon$ is a regularization term controlling the maximal multiplier on the learning rate, or equivalently the size of the trust region. The parameters $V_i \in \mathbb{R}^{N_i \times N_{i-1}}$ and $d_i \in \mathbb{R}^{N_i}$ are analogous to the canonical parameters of a neural network as introduced in Eq. 4, though operate in the space of whitened unit activations $U_i(h_i - c_i)$. This layer can be stacked to form a deep neural network having $L$ layers, with *model parameters* $\Omega = \{V_1, d_1, \cdots V_L, d_L\}$ and *whitening coefficients* $\Phi = \{U_0, c_0, \cdots, U_{L-1}, c_{L-1}\}$, as depicted in Fig. 1a.

Though the above layer might appear over-parametrized at first glance, we crucially *do not learn the whitening coefficients via loss minimization*, but instead estimate them directly from the model statistics. These coefficients are thus constants from the point of view of the optimizer and simply serve to improve conditioning of the Fisher with respect to the parameters $\Omega$, denoted $F_\Omega$. Indeed, using the same derivation that led to Eq. 5, we can see that the block-diagonal terms of $F_\Omega$ now involve terms $\mathbb{E} \left[ (U_i h_i)(U_i h_i)^T \right]$, which equals identity by construction.

### 3.2   Updating the Whitening Coefficients

As the whitened model parameters $\Omega$ evolve during training, so do the statistics $\mu_i$ and $\Sigma_i$. For our model to remain well conditioned, the whitening coefficients must be updated at regular intervals,

---

**Algorithm 1** Projected Natural Gradient Descent

---
1: **Input:** training set $\mathcal{D}$, initial parameters $\theta$.
2: **Hyper-parameters:** reparam. frequency $T$, number of samples $N_s$, regularization term $\epsilon$.
3: $U_i \leftarrow I; c_i \leftarrow 0; t \leftarrow 0$
4: **repeat**
5:     **if** $mod(t, T) = 0$ **then**                                  $\triangleright$ amortize cost of lines [6-11]
6:         **for** all layers $i$ **do**
7:             Compute canonical parameters $W_i = V_i U_{i-1}; b_i = d_i - W_i c_{i-1}$.    $\triangleright$ proj. $P_\Phi^{-1}(\Omega)$
8:             Estimate $\mu_i$ and $\Sigma_i$, using $N_s$ samples from $\mathcal{D}$.
9:             Update $c_i$ from $\mu_i$ and $U_i$ from eigen decomp. of $\Sigma_i + \epsilon I$.         $\triangleright$ update $\Phi$
10:           Update parameters $V_i \leftarrow W_i U_{i-1}^{-1}; d_i \leftarrow b_i + V_i U_{i-1} c_{i-1}$.        $\triangleright$ proj. $P_\Phi(\theta)$
11:         **end for**
12:     **end if**
13:     Perform SGD update wrt. $\Omega$ using samples from $\mathcal{D}$.
14:     $t \leftarrow t + 1$
15: **until** convergence

---

while taking care not to interfere with the convergence properties of gradient descent. This can be achieved by coupling updates to $\Phi$ with corresponding updates to $\Omega$ such that the overall function implemented by the MLP remains unchanged, e.g. by preserving the product $V_i U_{i-1}$ before and after each update to the whitening coefficients (with an analoguous constraint on the biases).

Unfortunately, while estimating the mean $\mu_i$ and $diag(\Sigma_i)$ could be performed online over a mini-batch of samples as in the recent Batch Normalization scheme [7], estimating the full covariance matrix will undoubtedly require a larger number of samples. While statistics could be accumulated online via an exponential moving average as in RMSprop [27] or K-FAC [8], the cost of the eigen-decomposition required for computing the whitening matrix $U_i$ remains cubic in the layer size.

In the simplest instantiation of our method, we exploit the smoothness of gradient descent by simply amortizing the cost of these operations over $T$ consecutive updates. SGD updates in the whitened model will be closely aligned to NGD immediately following the reparametrization. The quality of this approximation will degrade over time, until the subsequent reparametrization. The resulting algorithm is shown in the pseudo-code of Algorithm 1. We can improve upon this basic amortization scheme by updating the whitened parameters $\Omega$ using a per-batch diagonal natural gradient update, whose statistics are computed online. In our framework, this can be implemented via the reparametrization $W_i = V_i D_{i-1} U_{i-1}$, where $D_{i-1}$ is a diagonal matrix updated such that $\mathbb{V}[D_{i-1} U_{i-1} h_{i-1}] = 1$, for each minibatch. Updates to $D_{i-1}$ can be compensated for exactly and cheaply by scaling the rows of $U_{i-1}$ and columns of $V_i$ accordingly. A simpler implementation of this idea is to combine PRONG with batch-normalization, which we denote as PRONG$^+$.

### 3.3 Duality and Mirror Descent

There is an inherent duality between the parameters $\Omega$ of our whitened neural layer and the parameters $\theta$ of a canonical model. Indeed, there exist linear projections $P_\Phi(\theta)$ and $P_\Phi^{-1}(\Omega)$, which map from canonical parameters $\theta$ to whitened parameters $\Omega$, and vice-versa. $P_\phi(\theta)$ corresponds to line 10 of Algorithm 1, while $P_\Phi^{-1}(\Omega)$ corresponds to line 7. This duality between $\theta$ and $\Omega$ reveals a close connection between PRONG and Mirror Descent [3].

Mirror Descent (MD) is an online learning algorithm which generalizes the proximal form of gradient descent to the class of Bregman divergences $B_\psi(q, p)$, where $q, p \in \Gamma$ and $\psi : \Gamma \to \mathbb{R}$ is a strictly convex and differentiable function. Replacing the $L_2$ distance by $B_\psi$, mirror descent solves the proximal problem of Eq. 2 by applying first-order updates in a dual space and then projecting back onto the primal space. Defining $\Omega = \nabla_\theta \psi(\theta)$ and $\theta = \nabla_\Omega^* \psi(\Omega)$, with $\psi^*$ the complex conjugate of $\psi$, the mirror descent updates are given by:

$$\Omega^{(t+1)} = \nabla_\theta \psi\left(\theta^{(t)}\right) - \alpha^{(t)} \nabla_\theta \tag{8}$$

$$\theta^{(t+1)} = \nabla_\Omega \psi^*\left(\Omega^{(t+1)}\right) \tag{9}$$

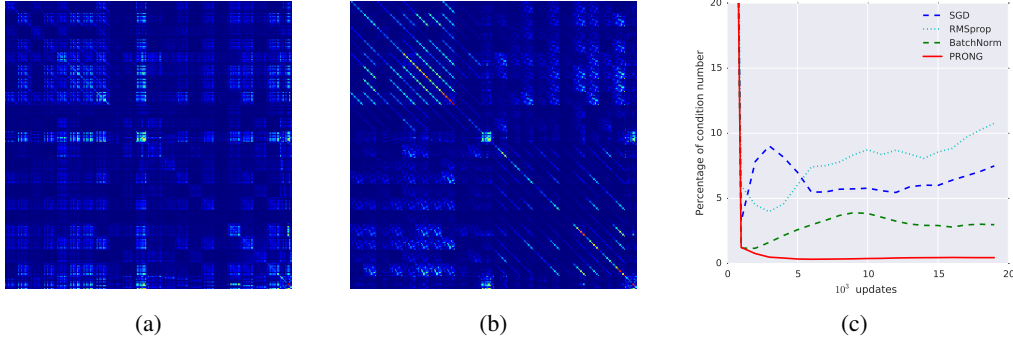

(a)            (b)            (c)

Figure 2: Fisher matrix for a small MLP (a) before and (b) after the first reparametrization. Best viewed in colour. (c) Condition number of the FIM during training, relative to the initial conditioning. All models where initialized such that the initial conditioning was the same, and learning rate where adjusted such that they reach roughly the same training error in the given time.

It is well known [26, 18] that the natural gradient is a special case of MD, where the distance generating function [1] is chosen to be $\psi(\theta) = \frac{1}{2}\theta^T F \theta$.

The mirror updates are somewhat unintuitive however. Why is the gradient $\nabla_\theta$ applied to the dual space if it has been computed in the space of parameters $\theta$? This is where PRONG relates to MD. It is trivial to show that using the function $\tilde{\psi}(\theta) = \frac{1}{2}\theta^T \sqrt{F}\theta$, instead of the previously defined $\psi(\theta)$, enables us to directly update the dual parameters using $\nabla_\Omega$, the gradient computed directly in the dual space. Indeed, the resulting updates can be shown to implement the natural gradient and are thus equivalent to the updates of Eq. 9 with the appropriate choice of $\psi(\theta)$:

$$
\begin{aligned}
\tilde{\Omega}^{(t+1)} &= \nabla_\theta \tilde{\psi}\left(\theta^{(t)}\right) - \alpha^{(t)}\nabla_\Omega = F^{\frac{1}{2}}\theta^{(t)} - \alpha^{(t)}\mathbb{E}_\pi\left[\frac{d\ell}{d\theta}F^{-\frac{1}{2}}\right] \\
\tilde{\theta}^{(t+1)} &= \nabla_\Omega \tilde{\psi}^*\left(\tilde{\Omega}^{(t+1)}\right) = \theta^{(t)} - \alpha^{(t)}F^{-1}\mathbb{E}_\pi\left[\frac{d\ell}{d\theta}\right]
\end{aligned}
\tag{10}
$$

The operators $\tilde{\nabla}\psi$ and $\tilde{\nabla}\psi^*$ correspond to the projections $P_\Phi(\theta)$ and $P_\Phi^{-1}(\Omega)$ used by PRONG to map from the canonical neural parameters $\theta$ to those of the whitened layers $\Omega$. As illustrated in Fig. 1b, the advantage of this whitened form of MD is that one may amortize the cost of the projections over several updates, as gradients can be computed directly in the dual parameter space.

### 3.4 Related Work

This work extends the recent contributions of [17] in formalizing many commonly used heuristics for training MLPs: the importance of zero-mean activations and gradients [10, 21], as well as the importance of normalized variances in the forward and backward passes [10, 21, 6]. More recently, Vatanen et al. [28] extended their previous work [17] by introducing a multiplicative constant $\gamma_i$ to the centered non-linearity. In contrast, we introduce a full whitening matrix $U_i$ and focus on whitening the feedforward network activations, instead of normalizing a geometric mean over units and gradient variances.

The recently introduced batch normalization (BN) scheme [7] quite closely resembles a *diagonal* version of PRONG, the main difference being that BN normalizes the variance of activations *before* the non-linearity, as opposed to normalizing the latent activations by looking at the full covariance. Furthermore, BN implements normalization by modifying the feed-forward computations thus requiring the method to backpropagate through the normalization operator. A diagonal version of PRONG also bares an interesting resemblance to RMSprop [27, 5], in that both normalization terms involve the square root of the FIM. An important distinction however is that PRONG applies this update in the whitened parameter space, thus preserving the natural gradient interpretation.

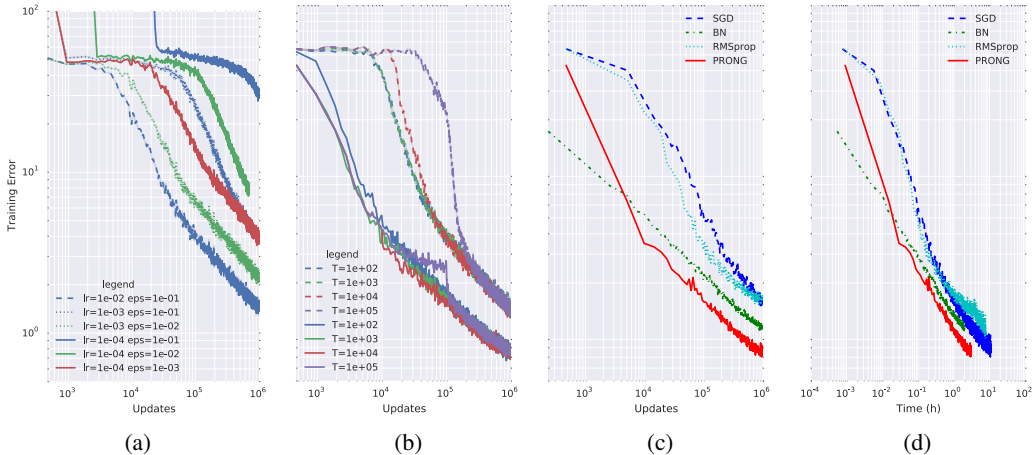

Figure 3: Optimizing a deep auto-encoder on MNIST. (a) Impact of eigenvalue regularization term $\epsilon$. (b) Impact of amortization period $T$ showing that initialization with the whitening reparametrization is important for achieving faster learning and better error rate. (c) Training error vs number of updates. (d) Training error vs cpu-time. Plots (c-d) show that PRONG achieves better error rate both in number of updates and wall clock.

K-FAC [8] is closely related to PRONG and was developed concurrently to our method. It targets the same layer-wise block-diagonal of the Fisher, approximating each block as in Eq. 5. Unlike our method however, KFAC does not approximate the covariance of backpropagated gradients as the identity, and further estimates the required statistics using exponential moving averages (unlike our approach based on amortization). Similar techniques can be found in the preconditioning of the Kaldi speech recognition toolkit [16]. By modeling the Fisher matrix as the covariance of a sparsely connected Gaussian graphical model, FANG [19] represents a general formalism for exploiting model structure to efficiently compute the natural gradient. One application to neural networks [8] is in decorrelating gradients across neighbouring layers.

A similar algorithm to PRONG was later found in [23], where it appeared simply as a thought experiment, but with no amortization or recourse for efficiently computing $F$.

## 4 Experiments

We begin with a set of diagnostic experiments which highlight the effectiveness of our method at improving conditioning. We also illustrate the impact of the hyper-parameters $T$ and $\epsilon$, controlling the frequency of the reparametrization and the size of the trust region. Section 4.2 evaluates PRONG on unsupervised learning problems, where models are both deep and fully connected. Section 4.3 then moves onto large convolutional models for image classification. Experimental details such as model architecture or hyper-parameter configurations can be found in the supplemental material.

### 4.1 Introspective Experiments

**Conditioning.** To provide a better understanding of the approximation made by PRONG, we train a small 3-layer MLP with tanh non-linearities, on a downsampled version of MNIST (10x10) [11]. The model size was chosen in order for the full Fisher to be tractable. Fig. 2(a-b) shows the FIM of the middle hidden layers before and after whitening the model activations (we took the absolute value of the entries to improve visibility). Fig. 2c depicts the evolution of the condition number of the FIM during training, measured as a percentage of its initial value (before the first whitening reparametrization in the case of PRONG). We present such curves for SGD, RMSprop, batch normalization and PRONG. The results clearly show that the reparametrization performed by PRONG improves conditioning (reduction of more than 95%). These observations confirm our initial assumption, namely that we can improve conditioning of the block diagonal Fisher by whitening activations alone.

**Sensitivity of Hyper-Parameters.** Figures 3a- 3b highlight the effect of the eigenvalue regularization term $\epsilon$ and the reparametrization interval $T$. The experiments were performed on the best

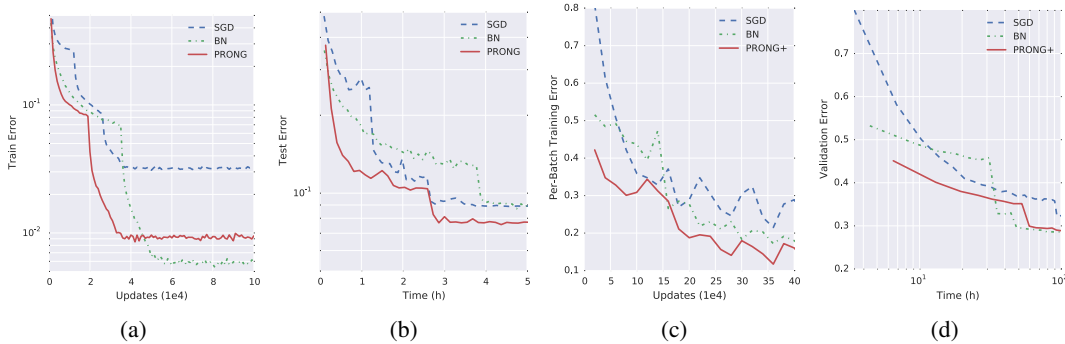

Figure 4: Classification error on CIFAR-10 (a-b) and ImageNet (c-d). On CIFAR-10, PRONG achieves better test error and converges faster. On ImageNet, PRONG$^+$ achieves comparable validation error while maintaining a faster covergence rate.

performing auto-encoder of Section 4.2 on the MNIST dataset. Figures 3a- 3b plot the reconstruction error on the training set for various values of $\epsilon$ and $T$. As $\epsilon$ determines a maximum multiplier on the learning rate, learning becomes extremely sensitive when this learning rate is high[2]. For smaller step sizes however, lowering $\epsilon$ can yield significant speedups often converging faster than simply using a larger learning rate. This confirms the importance of the manifold curvature for optimization (lower $\epsilon$ allows for different directions to be scaled drastically different according to their corresponding curvature). Fig 3b compares the impact of $T$ for models having a proper whitened initialization (solid lines), to models being initialized with a standard "fan-in" initialization (dashed lines) [10]. These results are quite surprising in showing the effectiveness of the whitening reparametrization as a simple initialization scheme. That being said, performance can degrade due to ill conditioning when $T$ becomes excessively large ($T = 10^5$).

## 4.2 Unsupervised Learning

Following Martens [12], we compare PRONG on the task of minimizing reconstruction error of a dense 8-layer auto-encoder on the MNIST dataset. Reconstruction error with respect to updates and wallclock time are shown in Fig. 3 (c,d). We can see that PRONG significantly outperforms the baseline methods, by up to an order of magnitude in number of updates. With respect to wallclock, our method significantly outperforms the baselines in terms of time taken to reach a certain error threshold, despite the fact that the runtime per epoch for PRONG was 3.2x that of SGD, compared to batch normalization (2.3x SGD) and RMSprop (9x SGD). Note that these timing numbers reflect performance under the optimal choice of hyper-parameters, which in the case of batch normalization yielded a batch size of 256, compared to 128 for all other methods. Further breaking down the performance, 34% of the runtime of PRONG was spent performing the whitening reparametrization, compared to 4% for estimating the per layer means and covariances. This confirms that amortization is paramount to the success of our method.[3]

## 4.3 Supervised Learning

We now evaluate our method for training deep supervised convolutional networks for object recognition. Following [7], we perform whitening across feature maps only: that is we treat pixels in a given feature map as independent samples. This allows us to implement the whitened neural layer as a sequence of two convolutions, where the first is by a 1x1 whitening filter. PRONG is compared to SGD, RMSprop and batch normalization, with each algorithm being accelerated via momentum. Results are presented on CIFAR-10 [9] and the ImageNet Challenge (ILSVRC12) datasets [20]. In both cases, learning rates were decreased using a "waterfall" annealing schedule, which divided the learning rate by 10 when the validation error failed to improve after a set number of evaluations.

**CIFAR-10** We now evaluate PRONG on CIFAR-10, using a deep convolutional model inspired by the VGG architecture [22]. The model was trained on $24 \times 24$ random crops with random horizontal reflections. Model selection was performed on a held-out validation set of 5k examples. Results are shown in Fig. 4. With respect to training error, PRONG and BN seem to offer similar speedups compared to SGD with momentum. Our hypothesis is that the benefits of PRONG are more pronounced for densely connected networks, where the number of units per layer is typically larger than the number of maps used in convolutional networks. Interestingly, PRONG generalized better, achieving $7.32\%$ test error vs. $8.22\%$ for batch normalization. This reflects the findings of [15], which showed how NGD can leverage unlabeled data for better generalization: the "unlabeled" data here comes from the extra crops and reflections observed when estimating the whitening matrices.

**ImageNet Challenge Dataset** Our final set of experiments aims to show the scalability of our method. We applied our natural gradient algorithm to the large-scale ILSVRC12 dataset (1.3M images labelled into 1000 categories) using the Inception architecture [7]. In order to scale to problems of this size, we parallelized our training loop so as to split the processing of a single minibatch (of size 256) across multiple GPUs. Note that PRONG can scale well in this setting, as the estimation of the mean and covariance parameters of each layer is also embarassingly parallel. Eight GPUs were used for computing gradients and estimating model statistics, though the eigen decomposition required for whitening was itself not parallelized in the current implementation. Given the difficulty of the task, we employed the enhanced version of the algorithm (PRONG+), as simple periodic whitening of the model proved to be unstable. Figure 4 (c-d) shows that batch normalisation and PRONG$^+$ converge to approximately the same top-1 validation error ($28.6\%$ vs $28.9\%$ respectively) for similar cpu-time. In comparison, SGD achieved a validation error of $32.1\%$. PRONG$^+$ however exhibits much faster convergence initially: after $10^5$ updates it obtains around $36\%$ error compared to $46\%$ for BN alone. We stress that the ImageNet results are somewhat preliminary. While our top-1 error is higher than reported in [7] ($25.2\%$), we used a much less extensive data augmentation pipeline. We are only beginning to explore what natural gradient methods may achieve on these large scale optimization problems and are encouraged by these initial findings.

## 5 Discussion

We began this paper by asking whether convergence speed could be improved by simple model reparametrizations, driven by the structure of the Fisher matrix. From a theoretical and experimental perspective, we have shown that Whitened Neural Networks can achieve this via a simple, scalable and efficient whitening reparametrization. They are however one of several possible instantiations of the concept of Natural Neural Networks. In a previous incarnation of the idea, we exploited a similar reparametrization to include whitening of backpropagated gradients[4]. We favor the simpler approach presented in this paper, as we generally found the alternative less stable for deep networks. This may be due to the difficulty in estimating gradient covariances in lower layers, a problem which seems to mirror the famous vanishing gradient problem. [17].

Maintaining whitened activations may also offer additional benefits from the point of view of model compression and generalization. By virtue of whitening, the projection $U_i h_i$ forms an ordered representation, having least and most significant bits. The sharp roll-off in the eigenspectrum of $\Sigma_i$ may explain why deep networks are ammenable to compression [2]. Similarly, one could envision spectral versions of Dropout [24] where the dropout probability is a function of the eigenvalues. Alternative ways of orthogonalizing the representation at each layer should also be explored, via alternate decompositions of $\Sigma_i$, or perhaps by exploiting the connection between linear auto-encoders and PCA. We also plan on pursuing the connection with Mirror Descent and further bridging the gap between deep learning and methods from online convex optimization.

**Acknowledgments**

We are extremely grateful to Shakir Mohamed for invaluable discussions and feedback in the preparation of this manuscript. We also thank Philip Thomas, Volodymyr Mnih, Raia Hadsell, Sergey Ioffe and Shane Legg for feedback on the paper.

## Footnotes

[1] As the Fisher and thus $\psi_\theta$ depend on the parameters $\theta^{(t)}$, these should be indexed with a time superscript, which we drop for clarity.

[2]Unstable combinations of learning rates and $\epsilon$ are omitted for clarity.

[3]We note that our whitening implementation is not optimized, as it does not take advantage of GPU acceleration. Runtime is therefore expected to improve as we move the eigen-decompositions to GPU.

[4]The weight matrix can be parametrized as $W_i = R_i^T V_i U_{i-1}$, with $R_i$ the whitening matrix for $\delta_i$.

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
