[Supplementary Material]

# Natural Neural Networks: Supplemental Material

**Guillaume Desjardins, Karen Simonyan, Razvan Pascanu, Koray Kavukcuoglu**
{gdesjardins,simonyan,razp,korayk}@google.com
Google DeepMind, London

This supplementary material provides the experimental details for Section 4.

## 4.2 Unsupervised Learning

The model consists of a dense 8-layer auto-encoder, trained to minimize reconstruction error on the MNIST dataset. The encoder is composed of 4 densely connected sigmoidal layers, with a number of hidden units per layer in $\{1\text{k}, 500, 250, 30\}$, and a symmetric (untied) decoder. Hyper-parameters were selected by grid search, based on training error, with the following grid specifications: training batch size in $\{32, 64, 128, 256\}$, fixed learning rates in $\{10^{-1}, 10^{-2}, 10^{-3}\}$ and momentum term in $\{0, 0.9\}$. For RMSprop, we further tuned the moving average coefficient in $\{0.99, 0.999\}$ and the regularization term controlling the maximum scaling factor in $\{0.1, 0.01\}$. For PRONG, we fixed the natural reparametrization to $T = 10^3$, using $N_s = 100$ samples (i.e. they were not optimized for wallclock time).

## 4.3 Supervised Learning

**CIFAR-10**    The model used for our CIFAR experiments consists of 8 convolutional layers, having $3 \times 3$ receptive fields. $2 \times 2$ spatial max-pooling was applied between stacks of two convolutional layers, with the exception of the last convolutional layer which computes the class scores and is followed by global max-pooling and soft-max non-linearity. This particular choice of architecture was inspired by the VGG model [1] and held fixed across all experiments. The number of filters per layer is as follows: $64, 64, 128, 128, 256, 256, 512, 10$.

Learning rates were decreased using a "waterfall" annealing schedule, which divided the learning rate by 10 when the validation error failed to improve by $1\%$ over 4 consecutive evaluations. Validation error was estimated every $10^3$ updates.

**ImageNet Challenge Dataset**    For all optimization algorithms, we considered initial learning rates in $\{10^{-1}, 10^{-2}, 10^{-3}\}$ and used a value of 0.9 as the momentum coefficient. For PRONG we tested reparametrization periods $T \in \{10, 10^2, 10^3, 10^4\}$, while typically using $N_s = 0.1T$. Eigenvalues were regularized by adding a small constant $\epsilon \in \{1, 10^{-1}, 10^{-2}, 10^{-3}\}$ before scaling the eigenvectors. Regularization consisted of a simple $L_2$ weight decay parameter of $10^{-4}$ with no Dropout [2]. Note that this grid was not searched exhaustively due to its prohibitive cost.

We again employed a "waterfall" schedule, which divided the learning rate by 10 if the validation error did not improve by $1\%$ after each epoch.