[Reviews · NeurIPS 2015]

Submitted by Assigned_Reviewer_1

The authors make this approach more efficient by whitening in block-diagonal fashion the weights of each layer, and by performing the whitening only periodically.

The most interesting section of this paper is the "Related Work", where they proceed to show many heuristic methods in NN optimization and in image processing are special cases or closely related to this method.

The paper does not present convincing evidence for the usefulness of the technique. There are comparisons to batch normalization (the state-of-the-art) on MNIST, CIFAR-10 and imageNet. On perhaps the most challenging dataset (ImageNet), this method did not show any improvement over BN. On the MNIST and CIFAR-10 it did show some modest improvement over BN.

It would also be of great interest to study approximation methods of the Fisher Matrix. While it O(n^3) to do eigen decomposition of the Fisher Matrix, for small matrices this shouldn't be a problem. Approaches such as the Nystrom Approximation can greatly speed up computation, while degrading the accuracy insignificantly.

Overall, I think this is an interesting approach that should be published at NIPS. Although the empirical results are fairly weak, I like the

fact that this method provides a theoretical justification for a set of heuristic methods, including Batch Normalization, and that together with the modest empirical results justifies publication.
Summary: This work proposes a theoretically sound approach to data normalization during training of deep convolutional networks. They illustrate speed ups in training while improving the accuracy of classification on CIFAR-10, and get close to state-of-the-art on ImageNet.

Submitted by Assigned_Reviewer_2

This paper develops a neural networks optimization method based on the idea of optimizing a version of the network where the values of the units at each layer are whitened and centered.

This is reminiscent of centering approaches, which only do the centering.

The method is presented as a type of approximate natural gradient descent, and competitive/superior performance compared to methods like SGD and Batch Normalization is demonstrated on various deep neural network training problems.

The paper is reasonably well written and the approximation approach seems interesting, although quite similar to several methods that have appeared recently (which are cited).

The empirical comparisons seems mostly solid, although I'm not convinced that they are as fair as they could be.

The approximation of the natural gradient used in the approach is not given any theoretical justification, although judging by the experiments it seems to be a reasonable one in practice.

L 35: what do you mean here by "probabilistic manifold" and "constant progress"?

Surely the amount of progress per iteration of natural gradient descent will change as optimization proceeds.

L 44: "do not take into account the smooth change".

Actually I think some of these methods initialize their inner iterations from previous solutions so they would benefit from a slowly changing Fisher

L 120:

Assuming this kind of independence seems like a very severe approximation to make, as these quantities are obviously closely related to each other.

Do you have any justification for making this approximation?

L 136:

PNGD seems like a poor choice for a name since it sounds like it could describe any method combining ideas of natural gradient descent with projected gradient descent, instead of the very specific approximations that you are using here (which are also neural-net specific).

L 152:

I don't see why this has anything obviously to do with a trust-region.

It should be spelled out in more detail.

L 191:

I found this paragraph very hard to follow.

I think you should be more explicit about what you mean here by this diagonal rescaling of U, providing an equation etc.

Are the rows of U being rescaled?

In this case wouldn't you then have to rescaling the columns of V to preserve the feed-forward computation?

Are the columns being rescaled (which is the interpretation I'm going with)?

Then I don't see how any simple manipulation of V would preserve the computation.

One could recompute V as W D^-1 U^-1, whenever U is replaced with U D, but this certainly doesn't correspond to a row scaling of V.

L 212: is the * supposed to be after \nabla like that?

L 215: Is it supposed to be \Omega^(t+1) in brackets here?

L 235: I don't understand what it means to use a "whitened mirror map".

Won't defining \psi completely change the kind of update you compute?

L 238: Won't the updates made to \Omega involve multiplying by inverses of matrices too?

How will this make things more efficient than the naive approach?

L 256: I think the square root connection is entirely spurious.

If what you do is equivalent to a form of natural gradient descent, then the updates, as considered in the original theta space, won't depend on square roots in any way.

This is much unlike the case with RMSprop.

L 260: I don't believe that the K-FAC approach uses any low-rank approximations.

L 310: An improved condition number doesn't imply that the matrix is close to diagonal.

Figure 2b) doesn't look particularly diagonal to me.

L 332: Why do plots c) and d) both have validation error on the y axis?

L 350: Sutskever et al (2013) showed that momentum is very important when optimizing these kinds of deep autoencoders and that the coefficient matters a lot.

Choosing a fixed value from {0,0.9} is probably causing SGD with momentum to underperform here.

L 384: What do you mean by extra perturbations?

Where is this coming from?

L 416: As far as I can tell, K-FAC does something like this but without the need for skip-connections, so it's not obvious that they should be needed.
Summary: This paper has some interesting ideas, despite similarities to some recent work on neural network optimization.

The experimental work seems reasonably well apart from a few issues.

Submitted by Assigned_Reviewer_3

This paper proposes an approximation to the natural gradient of DNN parameters by transforming the parameter to a space where the Fisher matrix is identity and so standard SGD approximates natural gradient descent. The presentation is clear and ties together several strands of work exploring similar ideas. The empirical results are convincing.

Some minor errors:

1. Line 106: h_{i-1}^T

instead of h_{i-t}^T

2. The formulas in section 3.1 and Algorithm 1 are not consistent. 2.a. Equation (6): should it be c_{i-1}?

2.b. Equation (6): either it is (h_{i-1} + c_{i-1}) and c_i = -\mu_i or line 7 in Algorithm 1 should be b_i = d_i - W_i c_{i-1} and c_i = \mu_i.

2.c. Algorithm 1 line 10: d_i \leftarrow b_i - W_i c_{i-1}, etc. Please make sure the formulas are consistent.

3. Line 256: bares -> bears
Summary: This paper studies improvements to convergence of DNN training by using an approximation to the natural gradients of the parameters. This is done by whitening the activations at each layer such that the Fisher matrix is identity. The paper is a nice addition to a growing body of work along similar directions.

Submitted by Assigned_Reviewer_4

A few questions/comments: 1) Can you add also batch normalization to Fig. 2c? What are the results there? 2) Can you run Fig. 3d and 4b for a few more hours (say, 10 more?)? It's not obvious that the errors in SGD (in Fig. 3d) and the BN network (in Fig. 4b) do not decrease further. What are the results there?

%%% Edit after author's feedback: Thank you for addressing my concerns. Please try to clarify your answer for 2 in the text.
Summary: Interesting method. However, it is not obvious that we get a significant improvement over batch normalization (See below).

Author Feedback
Author rebuttal: We thank the reviewers for their comments and suggestions. We will strive to address them in the final version, if accepted.

R1: reviewer summary
We want to clarify that we do not whiten the weights, but rather compute the gradients in a space where the activations at each layer are whiten, which approximately whitens the gradients on the weights. The forward computations in the model however stay unchanged (i.e. the output of each layer stays the same).

R1: "does not present convincing evidence"
While the final test error may not always improve (e.g. Imagenet), PNGD consistently improves convergence speed across all datasets. This improvement is especially pronounced for fully connected architectures. Figure 4 will also be modified to include training error on ImageNet, which shows PNGD reaching a better minima than BN.

R2 L35: "probabilistic manifold" - the term was coined by Amari (sometimes called neural manifold). It refers to the Riemannian manifold whose
metric is defined by the change in KL divergence over some parametric distribution; "constant progress" - natural gradient can be seen as taking a step in a descent direction such that the change in the density function (as measured by KL) is constant. See e.g. [24]

R2 L44: Agreed, we will soften this distinction.

R2 L120: The independence assumptions we make are clearly an approximation to the full Fisher, but one which aims to strike a balance between accuracy and runtime. We should keep in mind that SGD makes a much more severe approximation, i.e. that each unit is independent of other units in the same layer.

R2 L136: PNGD is meant to be generic in that it is a method for approximating NG via (amortized) SGD updates in a dual space. The nature of the projection certainly changes based on the approximation used. In our framework, these are part of the model however and the term "Whitened Neural Network" is meant to capture the specifics of per-layer whitening of activations. Perhaps results should refer to WNN-PNGD to reflect this.

R2 152: We will clarify this point. The connection is through the natural gradient, which can be considered a trust region method as it bounds the step size as a function of how much the model changes. Epsilon here plays a similar role to the size of the trust region, in that it ends up defining the maximal learning rate (for that layer).

R2 L191: Thank you for pointing this out. W = VD^-1DU, we simply set the new U = DU, and V = VD^-1. This amounts to rescaling the rows of U and the columns of V (and not rows as mentioned in the paper). We will clarify/correct this in the paper.

R2 L235: The whitened mirror map, which involves sqrt(F), is key to deriving an alternate form of Mirror Descent where dual parameters are updated using gradients computed in the dual space, a much more intuitive update rule in our opinion. We have modified the paper to make this derivation explicit.

R2 L238: Only \nabla_\omega \psi^* (projection P) requires computing matrix inverses. MD requires this projection at every gradient update due to the need for gradients to be computed in primal space. PNGD on the other hand can "stay" in the dual space over consecutive updates and thus amortize the cost of this projection. MD could conceivably be modified to use a fixed (over several updates) projection operator, however doing this efficiently would require similar techniques to those used in KFAC.

R2 L310: True, but an improved condition number does mean that the optimization is easier, as steps along the different eigen-directions have roughly the same impact.

R2 L332: One is plotted with respect to time while the other is wrt. number of updates. Replacing the third plot with "training error vs updates" would indeed be more informative and actually shows PNGD reaching a lower training error than BN.

R2 L350: This is indeed a stronger baseline and will be included.

R2 L384: This is a poor choice of words. "extra perturbations" refers to the "random crops with random horizontal reflections".

R2 L416: Indeed, gradients should technically be already centered in expectation due to the log loss being taken over a probability distribution. We will correct this statement.

R6: Can you add also batch normalization to Fig. 2c?
Thank you. We will provide BN results in the next revision.

R6: Longer runtime.
SGD actually starts to overfit at around 110 hours (with the validation error increasing slightly), while I can confirm that BN and PNGD+ both stop making progress on the validation error.